# Learngene Tells You How to Customize:
# Task-Aware Parameter Initialization at Flexible Scales

**Jiaze Xu** [1 2]  **Shiyu Xia** [1 2]  **Xu Yang** [1 2]  **Jiaqi Lv** [1 2]  **Xin Geng** [1 2]

## Abstract

Appropriate parameter initialization strategies are essential for reducing the high computational costs of training large pretrained models in various task scenarios. Graph HyperNetwork (GHN), a parameter initialization method, has recently demonstrated strong performance in initializing models. However, GHN still faces several challenges, including limited effectiveness in initializing larger models, poor performance on smaller datasets, and the requirement of task-specific GHN training, where each new task necessitates retraining the GHN model, leading to increased computational and storage overhead. To overcome these challenges, motivated by the recently proposed Learngene framework, we propose a novel method called *Task-Aware Learngene* (**TAL**). Briefly, our approach pretrains a TAL model under the guidance of a well-trained model and then performs multi-task tuning to obtain a shared TAL model that enables parameter prediction based on both model architectures and task-specific characteristics. Extensive experiments show the superiority of TAL. Models initialized with TAL outperform those initialized using GHN method by an average of 24.39% in terms of accuracy across Decathlon datasets. We provide the code at https://github.com/mathieuxu/Task-Aware-Learngene.

## 1. Introduction

Pretrained models have achieved remarkable success in computer vision (Dosovitskiy, 2020; Radford et al., 2021), natural language processing (Radford et al., 2019; Touvron et al., 2023), and other fields. However, pretraining such models requires computational resources and training costs (Radford et al., 2021; Touvron et al., 2023; Yang et al., 2024; Peng et al., 2025), thus making it challenging and costly to obtain well-trained models in many task scenarios with resource constraints (Mehta & Rastegari, 2021; Fu et al., 2025). So that an appropriate model initialization strategy becomes crucial, as effective parameter initialization not only accelerates model convergence but also enhances model performance (He et al., 2015; Zhang et al., 2018; Wang et al., 2022; 2024; Yao et al., 2025), thereby significantly reducing overall training costs.

Recently, Graph HyperNetwork (GHN) (Zhang et al., 2018; Knyazev et al., 2021; 2023) has been proposed as an approach to make model pretraining more accessible by reducing computational costs and enabling parameter initialization for models of multiple scales. Formally, given a set of neural network architectures $f$ as training data, where each architecture is represented as a computational graph $f^G$ (Knyazev et al., 2021), a GHN (denoted as $H_{\mathcal{D}}$ and parameterized by $\theta$) learns to predict the parameters of these neural networks. The prediction process can be formulated as $\mathbf{w}_{\text{pred}} = H_{\mathcal{D}}(f^G, \theta)$, where $\mathbf{w}_{\text{pred}}$ represents the predicted network parameters. During training, the GHN is optimized to minimize the loss function on a target dataset $\mathcal{D}$ like ImageNet-1K(Russakovsky et al., 2015). The predicted parameters $\mathbf{w}_{\text{pred}}$ demonstrate superior initialization performance compared to conventional random initialization methods, leading to reduced training time and computational costs.

Despite the advantages demonstrated by GHN, as illustrated in Fig.1(a), several significant limitations persist in their practical applications. First, although the latest GHN method LoGAH (Zhou et al., 2024) has demonstrated significant progress in handling deep neural architectures, its effectiveness diminishes when initializing larger models like ViT-base (Dosovitskiy, 2020). And most GHN methods still have considerable room for improvement in initialization accuracy across various datasets, such as ImageNet-1K(Russakovsky et al., 2015) and CIFAR-100 (Krizhevsky et al., 2009). Second, GHN requires independent training for each specific task, meaning that when facing different downstream applications, a new GHN model has to be retrained accordingly. This requirement not only substantially increases computational overhead but also imposes additional storage burdens.

[1]School of Computer Science and Engineering, Southeast University, Nanjing 210096, China [2]Key Laboratory of New Generation Artificial Intelligence Technology and Its Interdisciplinary Applications (Southeast University), Ministry of Education, China. Correspondence to: Jiaqi Lv <is.jiaqi.lv@gmail.com>.

*Proceedings of the $42^{nd}$ International Conference on Machine Learning*, Vancouver, Canada. PMLR 267, 2025. Copyright 2025 by the author(s).

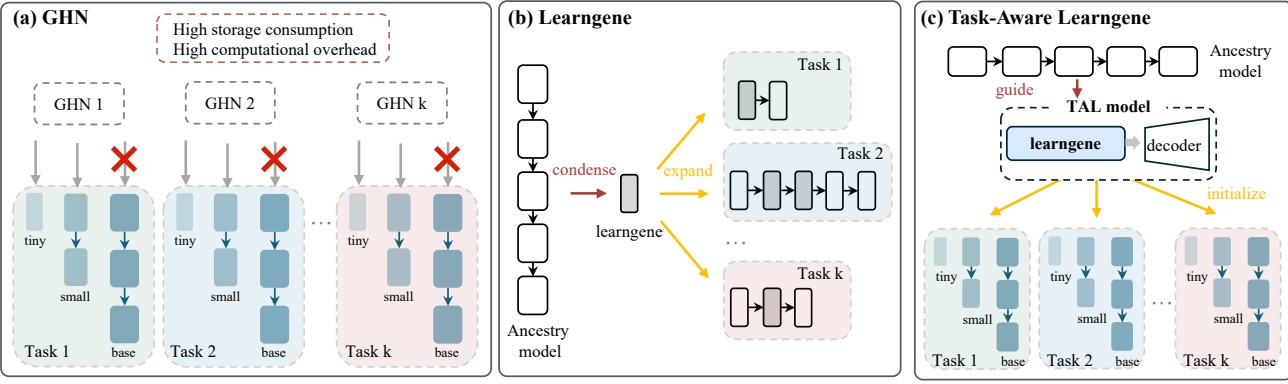

Figure 1: (a) GHN exhibits limitations in initializing large-scale models and requires high storage and computational resources for each individual task. (b) The Learngene framework condenses critical components (learngene) from a large ancestry model and expands them to initialize models of various sizes. (c) The Task-Aware Learngene (TAL) pipeline: first pretraining the TAL model using an ancestry model, then tuning it in a multi-task setting to generate task-specific initialization for models of different scales.

One recently proposed framework called Learngene (Wang et al., 2022), offers inspiring insights into addressing the limitations of GHN mentioned above. As shown in Fig.1(b), Learngene adopts a unique paradigm where it first condenses a well-trained model, termed ancestry model, into a small critical part known as learngene. Subsequently, this learngene can be expanded to initialize multiple models of varying sizes for different downstream tasks (Wang et al., 2022; Xia et al., 2024a; Shi et al., 2024; Feng et al., 2024b;a; Xie et al., 2024). While both Learngene and GHN aim to provide effective model initialization, Learngene distinguishes itself through its ability to inherit and utilize knowledge from the ancestry model and consider the commonalities across downstream tasks, enabling better adaptation to different application scenarios.

In this paper, we connect these two lines of work, proposing a novel approach called *Task-Aware Learngene* (**TAL**) that encodes shareable information to predict initial parameters for models across flexible scales. As shown in Fig.1(c), our approach involves two stages. First, we train the TAL model on a large dataset under the guidance of an ancestry model to transfer knowledge. Then, we tune the trained TAL model on multi-task datasets to effectively filter and convey the task-specific knowledge from the ancestry model. Finally, the trained TAL model can predicting model parameters for various tasks, even unseen ones and supports model customization at different scales. We systematically investigate the effectiveness of TAL. Extensive experiments show the superiority of TAL. For example, ViT-small model initialized by TAL achieving 22.44% higher accuracy on ImageNet-1K without any training compared to LoGAH (Zhou et al., 2024). Moreover models initialized with TAL also outperform those initialized using LoGAH by an average of 24.39% in terms

of accuracy across Decathlon datasets (Rebuffi et al., 2017).

The main contributions of this work are as follows:

(1) We design Task-Aware Learngene (TAL), an end-to-end mechanism that effectively represents and transfers shareable knowledge across tasks in parameter prediction. TAL not only provides well-initialized parameters for larger models but also enables parameter prediction based on both model architectures and task-specific characteristics.

(2) Extensive experiments demonstrate the superiority of TAL across various scenarios. Compared to training from scratch and previous GHN methods, models initialized with TAL achieve superior performance while substantially reducing both computational costs and storage requirements.

## 2. Related Work

### 2.1. Learngene

The Learngene method (Wang et al., 2022), inspired by biological gene inheritance, focuses on extracting compact components, known as learngene, from well pretrained models (ancestry models) to initialize models. Existing methods like Vanilla-LG (Wang et al., 2022), TLEG (Xia et al., 2024a), Learngene Pool (Shi et al., 2024) and SWS (Xia et al., 2024b) employ different strategies to select and expand learngene. Vanilla-LG extracts key layers as learngene and splices them with randomly initialized layers. TLEG uses linear expansion of learngene layers, while Learngene Pool refines large models into multiple small models, using their layers as learngene instances to construct new models. In Task-Aware Learngene (TAL), learngene becomes the encoder part of the TAL model rather than a sub block of the ancestry model. Model generation becomes an encoding-decoding process, with

ancestry model and multi-task knowledge injected through learngene. TAL can process both model and task information, initializing flexible-scale models for different tasks.

## 2.2. Graph HyperNetwork

Graph HyperNetwork (GHN) (Zhang et al., 2018; Knyazev et al., 2021) employs a hypernetwork for direct parameter prediction. This approach has attracted significant research interest due to its superior performance and remarkable adaptability. GHN-2 (Knyazev et al., 2021) and GHN-3 (Knyazev et al., 2023) further improved the parameter prediction capabilities of GHN by improving the learning process of the model computation graph. The latest LoGAH method (Zhou et al., 2024) introduces low-rank approximation (LoRA) technology, allowing GHN to predict the parameters of larger models using smaller hypernetworks. This progress has greatly improved the efficiency and ability of GHN in handling large-scale model parameter prediction tasks. Our task-aware learngene (TAL) incorporates modules to process task information, enabling a single TAL model to customize models of varying scales for different tasks.

## 3. Background

A Graph HyperNetwork (GHN) is a neural network $H_{\mathcal{D}}$ parameterized by $\theta$ and trained on a dataset $\mathcal{D}$. The input of GHN $H_D(\theta)$ is a computational graph $f^G$ of a neural network $f$ and the output of GHN is the parameters of the model $\mathbf{w}_{\text{pred}} = H_D(f^G; \theta)$.

In (Knyazev et al., 2021), GHN $H_D$ is trained by SGD over $M$ training architectures $\{f_a^G\}_{a=1}^M$ and $N$ training data samples $\{x_j, y_j\}_{j=1}^N$ on the following optimization problem:

$$\underset{\theta}{\arg\min} \frac{1}{NM} \sum_{j=1}^{N} \sum_{a=1}^{M} \mathcal{L}(f_a(x_j; H_D(f_a^G; \theta)), y_j), \quad (1)$$

when training GHN $H_D(\theta)$, a meta-batch of $m$ training architectures is sampled as input for GHN. Meanwhile, a mini-batch of $n$ training datas $\mathbf{x}$ is sampled and fed into the parameter-predicted $m$ architectures to get $m \times n$ predictions $\hat{y}$. The cross-entropy loss $\mathcal{L}$ is computed between $\hat{y}$ and ground truth labels $y$ of $\mathbf{x}$ for classification tasks. Afterward, the loss is back-propagated to update the parameters $\theta$ of $H_D$ by gradient descent.

The input computational graph $f^G = (V, E)$ is a directed acyclic graph (DAG), where the nodes $V$ correspond to operations (convolution, pooling, self-attention, etc.) (Knyazev et al., 2021), while the edges $E$ correspond to the forward pass flow of inputs through the network $f$. GHN takes $d$-dimensional node features $\mathbf{H}^{(1)} \in \mathbb{R}^{|V| \times d}$ as input obtained using an embedding layer for each $i$-th node:

$\mathbf{h}_i^{(1)} = \text{Embed}(\mathbf{h}_i^{(0)})$, where $\mathbf{h}_i^{(0)}$ is a one-hot vector denoting an operation.

## 4. Task-Aware Learngene

Fig.2 (a-c) illustrates the overall pipeline of Task-Aware Learngene (TAL). First, we train the TAL model on a large dataset under the guidance of an ancestry model to transfer knowledge. Then, we tune the trained TAL model on multi-task datasets to effectively filter and convey the task-specific knowledge from the ancestry model to downstream models cross different tasks. Finally, the trained TAL model can predicting model parameters for various tasks, even unseen ones and supports model customization at different scales.

**TAL model structure and components.** In the TAL, we adopt encoder-decoder structure for model parameters prediction (Knyazev et al., 2023; Zhou et al., 2024). We refer to the encoder part of the TAL model as learngene because it first inherits knowledge from the ancestry model and then transfers task-specific knowledge based on different tasks. Specifically, learngene receives both model configuration through model computational graph and task information from ancestry model. Based on task information, learngene can filter out task-specific knowledge which previously inherited from the ancestry model and inject it into model computational graphs, thereby producing task-specific computational graphs.

The architecture of learngene is shown in Fig.2(d). Inspired by (Perez et al., 2018; Oreshkin et al., 2018), we introduce a task hypernet $h$ that processes task information to dynamically generate parameters for the task-specific layer, which is implemented as a simple MLP. Then task-specific layer acts on the model computational graph, transferring task information to it.

In this process, task information is passed in the form of a task embedding $\{I_\tau\}_{\tau=1}^T$ for each task, which is generated by the ancestry model through the average feature extraction of the task images (Vu et al., 2020).

The task hypernet $h$ generates task bias parameters $\gamma_\tau$, and $\beta_\tau$ of the task-specific layer.

$$(\gamma_\tau, \beta_\tau) := h(I_\tau) = (W^\gamma, W^\beta) I_\tau, \quad (2)$$

where $W^\gamma \in \mathbb{R}^{h \times t}$ and $W^\beta \in \mathbb{R}^{h \times t}$.

The task-specific layers apply these bias parameters to the model's computation graph using the following formula:

$$f_\tau^G = \gamma_\tau \times f^G + \beta_\tau \quad (3)$$

The task-specific model computational graph generated by learngene is then passed to the decoder (Zhou et al., 2024), which decodes the graph to generate the model parameters.

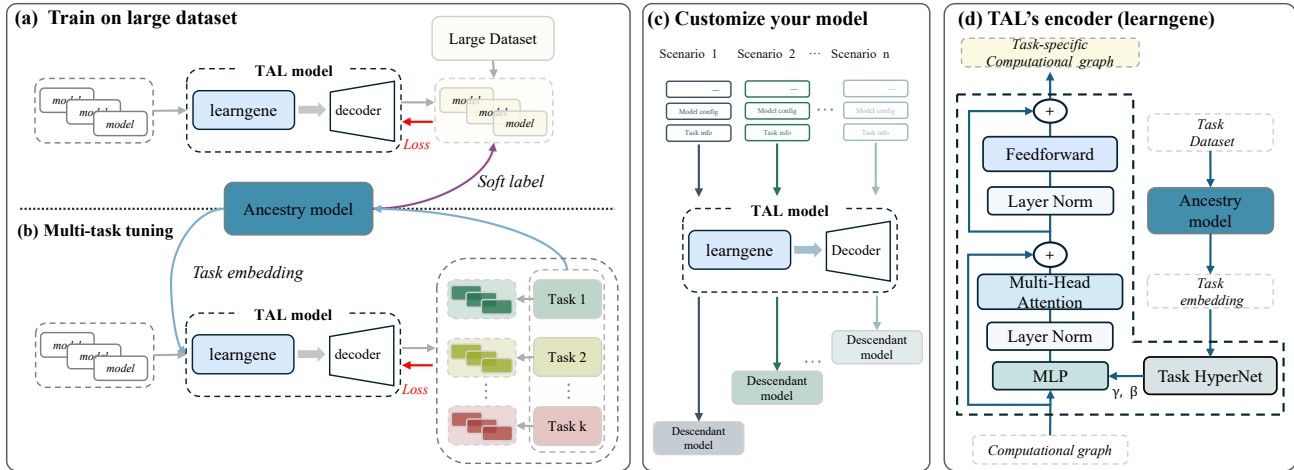

Figure 2: (a) Training TAL model on a large dataset under the guidance of a large-scale foundation model (ancestry model). (b) Tuning TAL model to multiple tasks. (c) Customizing task-specific models with flexible scale based on new task scenarios. (d) The learngene is based on a transformer architecture and consists of a stack of transformer blocks.

**Train TAL model on a large dataset.** In order to inherit the knowledge of the ancestry model, we first train the TAL model on a large dataset under the guidance of the ancestry model. We adopt the method from (Shu et al., 2021), converting features extracted by the ancestry model from the images into a probability distribution map. For the training models of the TAL, we also apply the same method to obtain their feature probability distributions and compute the KL divergence between those of the ancestry model. This function is denoted as:

$$\mathcal{L}_{\text{aux}} = \text{KL}(\text{softmax}(E_{\text{train}}M) || \text{softmax}(E_{\text{anc}})), \quad (4)$$

where $E_{\text{train}}$ and $E_{\text{anc}}$ refer to the encoders' output of the training model and ancestry model respectively. The matrix $M \in \mathbb{R}^{d \times d'}$, which transforms the output dimension $d$ of $E_{\text{train}}$ to match the output dimension $d'$ of $E_{\text{anc}}$, like the parameters of other parts of the training model, the transformation matrix's parameters are predicted directly by TAL model.

Considering loss between training model's predicted label and ground truth label:

$$\mathcal{L}_{cls} = \text{CE}(y_c, f_{\text{train}}(x)), \quad (5)$$

where $f_{\text{train}}(x)$ represents the training model's predicted label of input image data $x$ and $y_c$ denotes the ground truth label belonging to category c. Then, while one model is used as training data for TAL model, the total training loss is computed as follows:

$$\mathcal{L} = \alpha\mathcal{L}_{aux} + (1-\alpha)\mathcal{L}_{cls}, \quad (6)$$

Through this process, the TAL model can inherit and utilize the vast amount of domain knowledge already learned in the ancestry model, enabling models initialized by TAL model to handle complex tasks.

**Tuning TAL model under multi-task setting.** We then tune the TAL model on multiple tasks, leveraging task information to enable learngene to generate task-specific computation graphs, thereby decoding the model parameters tailored to each task. We formulate the loss function for this part of the TAL model's training. Given the data from a set of tasks $\{\mathcal{D}_\tau\}_{\tau=1}^T$, here $T$ is the total number of tasks and $\mathcal{D}_\tau = \{(x_i^\tau, y_i^\tau)\}_{i=1}^{N_\tau}$ shows the training data for $\tau$-th task with $N_\tau$ samples.

Assuming there is a TAL model $H_D(\theta)$ parameterized by $\theta$ that computes the output the parameters of the model $\mathbf{w}_{\text{pred}} = H_D(f^G; \theta)$ for input computational graph $f^G$ of a neural network. In multi-task setting, TAL model is trained by SGD over $M$ training models $\{f_a^G\}_{a=1}^M$ and T training tasks $\{\mathcal{D}_\tau\}_{\tau=1}^T$ on the following optimization problem:

$$\operatorname*{argmin}_\theta \frac{1}{TM}\sum_{\tau=1}^T\sum_{a=1}^M\sum_{(x_\tau^j, y_\tau^j) \in \mathcal{D}_\tau} w_\tau \mathcal{L}(f_a(x_\tau^j; H_D(f_a^G; \theta)), y_\tau^j),$$

$$(7)$$

where $\mathcal{L}$ is typically the cross-entropy loss and $w_\tau$ shows the sampling weight for $\tau$-th task.

Multi-task training allows the models predicted by TAL to inherit task-specific knowledge filtered by learngene from the ancestry model, as well as the shared knowledge across tasks.

**Customize models across different tasks.** After training on a large dataset and tuning on multiple tasks, the TAL model can provide task-specific, variable-sized models for both seen and unseen tasks. By simply passing the required model configuration and task information to the TAL model,

Table 1: Performance of models on ImageNet-1K initialized with GHN-3, LoGAH, TAL and TAL[+], after 75 epochs of training for all initialization methods.

| MODEL | TRAINING STATE | GHN-3 | LoGAH | TAL | TAL[+] |
|---|---|---|---|---|---|
| 12-TINY | UNTRAINED | **34.45** | 22.79 | 26.20 | 31.31 |
| | TRAINED | 48.93 | 62.53 | **63.03** | 60.04 |
| 12–SMALL | UNTRAINED | 31.03 | 16.78 | 23.28 | **39.22** |
| | TRAINED | 53.47 | 65.41 | **66.61** | 65.63 |
| 12-BASE | UNTRAINED | 0.10 | 0.10 | 0.10 | **38.72** |

one can instantly obtain well-initialized model parameters tailored to the task at hand.

**Model architecture datasets.** In GHN-1/2 (Knyazev et al., 2021) and GHN-3 (Knyazev et al., 2023), training architectures are sampled from DeepNets-1M, a dataset of 1 million architectures (Knyazev et al., 2021). In LoGAH works, they constructed task-specific architecture datasets: ViTs-1K for vision tasks and GPTs-1K for language tasks (Zhou et al., 2024), each containing 1K different computational graphs. For our TAL method, we adopt these datasets from LoGAH, and additionally design an enhanced vision model library denoted as ViTs[+]-1K dataset for vision tasks. Unlike the original ViTs-1K which has a 10M parameter constraint, our ViTs[+]-1K incorporates wider and deeper model architectures. We also increase the proportion of larger models in the dataset. This design strategy leads to significant improvements in TAL's performance. The model trained with this enhanced library is denoted as TAL[+].

## 5. Experiments

In this section, we evaluate our proposed TAL method by predicting parameters for models of various scales across different tasks, comparing it with previous methods including GHN-3, LoGAH, and random initialization. First, we assess TAL's capability in predicting ViT model parameters on both seen and unseen visual tasks during training. Then, we examine TAL's effectiveness in predicting parameters for GPT-2 models on natural language tasks. Furthermore, we conduct comprehensive ablation studies to investigate the impact of various factors on TAL's performance. Finally, we visualize the intermediate learngene outputs to demonstrate TAL's effectiveness in handling multiple tasks.

### 5.1. Experimental Setup

**Datasets.** Our experiments comprehensively evaluate our proposed methods (TAL and TAL[+]) against existing approaches (GHN-3 and LoGAH) on both vision and language tasks. Each experiment requires two types of datasets: model architecture datasets for parameter prediction and task-specific datasets for downstream evaluation.

For vision experiments, we employ two model architecture

datasets: the ViTs-1K dataset (Zhou et al., 2024) for TAL, GHN-3, and LoGAH methods, and the ViTs[+]-1K dataset for our enhanced TAL[+] method. Both datasets contain 1000 different ViT-style computational graphs as the architecture source. The vision tasks are evaluated on the Visual Domain Decathlon Challenge (Rebuffi et al., 2017), which comprises 10 diverse datasets: (1) ImageNet-1K (**IN-1K**)(Russakovsky et al., 2015), (2) CIFAR-100 (**C100**)(Krizhevsky et al., 2009), (3) Aircraft (**Airc.**)(Maji et al., 2013), (4) Daimler pedestrian classification (**DPed**)(Munder & Gavrila, 2006), (5) Describable textures (**DTD**)(Cimpoi et al., 2014), (6) German traffic signs (**GSTR**)(Stallkamp et al., 2012), (7) Omniglot (**OGlt**)(Lake et al., 2015), (8) **SVHN**(Netzer et al., 2011), (9) UCF101 Dynamic Images (**UCF**)(Soomro et al., 2012), and (10) Flowers102 (**Flwr**)(Nilsback & Zisserman, 2008).For detailed dataset descriptions, please refer to Appendix .1.

For language experiments, all methods utilize GPTS-1K (Zhou et al., 2024) as the model architecture dataset. The language tasks are evaluated on four widely-used NLP benchmarks: Microsoft Research Paraphrase Corpus (**MRPC**) (Dolan & Brockett, 2005), Corpus of Linguistic Acceptability (**CoLA**) (Warstadt, 2019), Recognizing Textual Entailment (**RTE**) (Wang, 2018), and Internet Movie Database reviews (**IMDB**) (Maas et al., 2011).

**Baselines.** We compare TAL with GHN-3 (Knyazev et al., 2023) and the latest LoGAH method (Zhou et al., 2024), which improves the design of the decoder and significantly enhances the initialized models' performance. To ensure fair comparison, we reproduce all experiments using the official source code of these methods under identical experimental settings and environment.

**Sampling tasks.** During multi-task training, we sample tasks using conventional temperature-based sampling (Raffel et al., 2020) with a temperature of $T = 2$ for all methods. Tasks are sampled proportionally to $p_\tau^{1/T}$, where $p_\tau = \frac{N_\tau}{\sum_{i=1}^T N_i}$ and $N_\tau$ represents the number of training samples for the $\tau$-th task. Note that this sampling probability $p_\tau^{1/T}$ directly corresponds to the sampling weight $w_\tau$ introduced in Formula 7.

**Training Details.** For both TAL and TAL[+], we first pretrain the hypernets on ImageNet-1K for 75 epochs, followed by 100 epochs of multi-task training on the Decathlon Challenge datasets. For TAL[+], we leverage the logits from the ancestry model as soft labels to guide the training process. All models are trained using automatic mixed precision in PyTorch, with a cosine annealing learning rate schedule starting at $lr = 3e-4$, weight decay $\lambda = 1e-2$ and predicted parameter regularization $\gamma = 3e-5$ (Knyazev et al., 2023). We use a pretrained ViT-Base (Dosovitskiy, 2020) as the ancestry model.

Table 2: Performance of untrained models on Decathlon tasks initialized with GHN-3, LoGAH, TAL and TAL$^+$. Note that for ViT-base, only TAL$^+$ initialization results are shown, as it is the only method capable of predicting parameters for base-scale models.

| MODEL | METHOD | AIRC. | C100 | DPED | DTD | GSTR | OGLE | SVHN | UCF | FLWR |
|---|---|---|---|---|---|---|---|---|---|---|
| 3-TINY | GHN-3 | 3.12 | 34.06 | 85.41 | 6.38 | 87,77 | 0.06 | 10.00 | 2.25 | 7.06 |
| | LoGAH | 2.58 | 29.16 | 78.83 | 8.30 | 92.12 | 0.26 | 17.32 | 3.94 | 6.96 |
| | TAL | **6.69** | 39.53 | 79.83 | 22.55 | 94.86 | **28.37** | 83.31 | 28.33 | 33.82 |
| | TAL$^+$ | 2.04 | **50.80** | **88.88** | **29.26** | **98.87** | 0.11 | **87.37** | **35.76** | **50.00** |
| 6-TINY | GHN-3 | 3.24 | 35.19 | 87.72 | 6.86 | 89.12 | 0.06 | 10.00 | 3.43 | 10.88 |
| | LoGAH | 3.21 | 46.33 | 81.19 | 9.20 | 96.82 | 0.31 | 20.82 | 5.02 | 8.33 |
| | TAL | **17.43** | 48.95 | 85.87 | 28.30 | **99.25** | 48.98 | 89.85 | **38.63** | 45.69 |
| | TAL$^+$ | 1.98 | **51.01** | **88.06** | 28.62 | 98.89 | 0.18 | 87.27 | 35.71 | **49.71** |
| 12-TINY | GHN-3 | 3.15 | 31.12 | 85.63 | 6.86 | 85.02 | 0.06 | 10.00 | 3.07 | 9.90 |
| | LoGAH | 3.15 | 33.80 | 79.86 | 9.41 | 96.52 | 0.18 | 20.56 | 5.53 | 9.12 |
| | TAL | **16.71** | 44.72 | 84.76 | 28.03 | **99.15** | 28.11 | 88.87 | **39.81** | 45.00 |
| | TAL$^+$ | 2.07 | **50.99** | **88.21** | **29.41** | 98.94 | 0.14 | 87.30 | 35.81 | **49.80** |
| 3-SMALL | GHN-3 | 2.91 | 35.75 | 86.77 | 6.70 | 87.68 | 0.06 | 10.00 | 2.61 | 10.39 |
| | LoGAH | 3.15 | 44.95 | 80.66 | 9.36 | 96.35 | 0.28 | 19.91 | 5.33 | 8.73 |
| | TAL | **17.16** | 47.25 | 83.78 | 27.23 | 98.93 | **47.75** | 87.83 | 38.32 | 44.31 |
| | TAL$^+$ | 1.95 | **56.05** | **94.20** | **33.14** | **99.57** | 0.03 | **91.24** | **47.23** | **55.78** |
| 6-SMALL | GHN-3 | 3.12 | 35.30 | 86.80 | 7.34 | 90.05 | 0.06 | 10.00 | 2.36 | 12.75 |
| | LoGAH | 3.18 | 45.93 | 80.60 | 9.95 | 96.97 | 0.26 | 21.04 | 5.38 | 8.63 |
| | TAL | **17.85** | 49.88 | 83.91 | 28.67 | 99.30 | **50.29** | 89.82 | 40.83 | 46.47 |
| | TAL$^+$ | 1.98 | **56.08** | **94.42** | **32.98** | **99.62** | 0.09 | **91.31** | **48.16** | **56.57** |
| 12-SMALL | GHN-3 | 2.64 | 5.55 | 84.30 | 7.23 | 84.39 | 0.06 | 10.00 | 1.74 | 9.51 |
| | LoGAH | 2.64 | 35.77 | 80.39 | 8.24 | 96.81 | 0.18 | 20.66 | 4.30 | 6.57 |
| | TAL | **17.16** | 45.63 | 82.16 | 27.87 | 99.18 | **39.66** | 88.55 | 40.98 | 45.10 |
| | TAL$^+$ | 1.86 | **56.04** | **94.66** | **33.14** | **99.58** | 0.14 | **91.30** | **48.21** | **56.57** |
| 3-BASE | TAL$^+$ | 2.22 | 55.99 | 94.10 | 32.93 | 99.41 | 0.09 | 90.99 | 4.39 | 56.08 |
| 6-BASE | TAL$^+$ | 1.80 | 56.02 | 94.00 | 33.03 | 99.41 | 0.11 | 91.05 | 47.18 | 56.08 |
| 12-BASE | TAL$^+$ | 2.31 | 56.06 | 94.39 | 32.93 | 99.41 | 0.11 | 90.97 | 47.59 | 55.69 |

## 5.2. Main results

**TAL/ TAL$^+$ achieves better performance on ImageNet-1K.** We evaluate the performance of the TAL/TAL$^+$ on the ImageNet-1K. As shown in Tab.1, the untrained model, structured as ViT-Small and initialized using TAL$^+$, outperforms it initialized with LoGAH by **22.44%** on ImageNet-1K. Furthermore, after 75 epochs training, the model initialized with TAL achieves **1.20%** higher accuracy compared to LoGAH initialization. Notably, among all initialization methods, only TAL$^+$ is capable of predicting parameters for ViT-Base scale models, achieving an initialization accuracy of **38.72%**. These results show that TAL can effectively inherit and utilize the knowledge already learned in the ancestry model.

**Models initialized with TAL/ TAL$^+$ demonstrate strong performance without any training on Decathlon tasks.** We compare TAL/ TAL$^+$ with GHN-3 and LoGAH methods on Decathlon tasks using ViT models of varying architectures and depths. The experiments are conducted with ViT-Tiny and ViT-Small at three different depths: 3, 6, and 12 layers, as well as ViT-Base models. Notably, for ViT-Base

architectures, only TAL$^+$ results are presented since it is the only method capable of parameter prediction for base-scale models. As shown in Tab.2, untrained models initialized with TAL/ TAL$^+$ outperform the GHN-3 and LoGAH across all Decathlon tasks.

**Models initialized with TAL/TAL$^+$ outperform those initialized by the LoGAH method during the training process.** We select 12-layer ViT-tiny (12-Tiny) and a 12-layer ViT-small (12-Small) as test models for further evaluation. Our TAL/TAL$^+$ initialization consistently outperforms other initialization methods across all tasks. Notably, on DTD, UCF, and Flower datasets, models initialized with our method achieve approximately **15-25%** higher accuracy after training compared to other initialization approaches.

**The TAL method significantly reduces computational costs.** We calculate the total training time for each method across the previous ten experimental datasets. As shown in Tab.4, under identical experimental conditions, the TAL method demonstrates notable efficiency advantages, reducing training time by **22%** compared to the LoGAH

Table 3: Performance of trained models on Decathlon tasks initialized with RandInit, GHN-3, LoGAH, TAL and TAL$^+$. For models initialized with RandInit, accuracy is reported after 200 epochs of training for each task, while for models initialized with other methods, trained for 100 epochs.

| MODEL | METHOD | AIRC. | C100 | DPED | DTD | GSTR | OGLE | SVHN | UCF | FLWR |
|---|---|---|---|---|---|---|---|---|---|---|
| 12-TINY | RANDINIT | 7.80 | 58.64 | 98.74 | 23.09 | 99.58 | 24.12 | 86.38 | 24.03 | 35.10 |
| | GHN-3 | 4.80 | 55.88 | 98.20 | 20.11 | 97.53 | 9.77 | 83.08 | 24.80 | 32.45 |
| | LOGAH | 5.61 | 58.93 | 97.47 | 20.32 | 99.23 | 29.22 | 86.13 | 35.81 | 35.78 |
| | TAL | **19.17** | **59.80** | 98.88 | 29.73 | 99.60 | **57.50** | 90.31 | 46.47 | 48.33 |
| | TAL$^+$ | 7.23 | 58.90 | **99.46** | **33.09** | **99.78** | 24.01 | **90.94** | **49.28** | **58.04** |
| 12-SMALL | RANDINIT | 8.01 | 60.17 | 98.44 | 25.05 | 98.89 | 15.70 | 85.57 | 23.31 | 35.00 |
| | GHN-3 | 5.22 | 57.35 | 98.32 | 15.21 | 97.36 | 20.98 | 80.21 | 22.34 | 34.51 |
| | LOGAH | 7.20 | 59.98 | 97.45 | 20.21 | 98.95 | 24.55 | 85.10 | 33.86 | 34.51 |
| | TAL | **18.69** | 60.49 | 98.89 | 31.01 | 99.77 | **57.01** | **92.02** | 47.75 | 49.71 |
| | TAL$^+$ | 8.61 | **60.77** | **99.51** | **34.36** | **99.79** | 23.04 | 91.72 | **52.77** | **59.02** |

Table 4: Training time comparison of different methods, all experiments run on an NVIDIA RTX 4090 with time measured in hours (h).

| METHOD | GHN-3 | LOGAH | TAL | TAL$^+$ |
|---|---|---|---|---|
| TIME(HOURS) | 58.81 | 46.55 | **36.19** | 70.53 |

Table 5: Performance of models on unseen tasks initialized with RandInit, LoGAH and TAL, after 5 and 100 epochs of training for each task.

| DATASET | MODEL | EPOCHS | RANDINIT | LOGAH | TAL |
|---|---|---|---|---|---|
| F-MNIST | 3-TINY | 5 | 82.71 | 87.56 | **88.86** |
| | | 100 | 89.41 | **91.08** | 90.99 |
| | 6-SMALL | 5 | 83.66 | 87.78 | **89.78** |
| | | 100 | 88.98 | 90.68 | **91.56** |
| FER2013 | 3-TINY | 5 | 29.20 | 42.57 | **47.06** |
| | | 100 | 59.91 | 60.60 | **61.41** |
| | 6-SMALL | 5 | 30.00 | 32.57 | **49.99** |
| | | 100 | 62.55 | 61.94 | **65.09** |
| HAM10000 | 3-TINY | 5 | 83.09 | **87.56** | 86.59 |
| | | 100 | 97.46 | **97.71** | **97.71** |
| | 6-SMALL | 5 | 82.13 | 89.61 | **91.18** |
| | | 100 | 97.70 | 97.83 | **97.95** |

Table 6: Performance of untrained GPT2 models on NLP tasks initialized with LoGAH and TAL.

| MODEL | METHOD | MRPC | COLA | RTE | IMDB |
|---|---|---|---|---|---|
| 3-GPT2 | LOGAH | 55.88/ 68.75 | 0.70 | 47.65 | 52.06 |
| | TAL | 62.99/ 76.70 | 2.89 | 46.21 | 63.21 |
| 6-GPT2 | LOGAH | 59.07/ 72.12 | 1.23 | 46.21 | 53.58 |
| | TAL | 61.52/ 73.70 | 3.32 | 47.29 | 62.76 |
| 9-GPT2 | LOGAH | 60.78/ 74.19 | 1.23 | 48.01 | 57.76 |
| | TAL | 58.09/ 68.97 | 1.75 | 48.78 | 62.09 |
| 12-GPT2 | LOGAH | 56.13/ 68.98 | 0.10 | 48.01 | 58.07 |
| | TAL | 52.70/ 60.53 | 3.05 | 49.82 | 61.49 |
| AVG_ACC | LOGAH | 57.96/ **71.01** | 0.82 | 47.47 | 55.37 |
| | TAL | **58.83**/ 69.98 | **2.75** | **48.02** | **62.39** |

the classification of skin lesions. We select 3-layer ViT-tiny (3-Tiny) and a 6-layer ViT-small (6-Small) as test models for further evaluation. Tab.5 shows that models initialized with TAL converge faster and achieve higher test accuracy on unseen downstream tasks.

**TAL shows promising results on NLP tasks.** We further evaluate TAL's effectiveness on NLP tasks by conducting experiments on MRPC, COLA, RTE and IMDB datasets using GPT2-small models with 3, 6, 9, and 12 layers. For training setup, we individually train a separate LoGAH model for each task with 250 epochs, while in TAL method, we leverage pretrained GPT-2 model (Radford et al., 2019) to provide task information and train a single shared TAL model across all four tasks with only 100 epochs. Compared to LoGAH, TAL not only reduces the training time by more than 50% but also demonstrates competitive or superior performance across test tasks. Tab.6 shows TAL's notable improvement on COLA and IMDB tasks, with accuracy gains of **1.93%** and **7.02%** respectively on average.

### 5.3. Analysis and Ablation

In the main experiments, high-quality model initialization is shown to significantly accelerate convergence and improve

method. Although TAL$^+$ requires a longer training time (70.53 hours), this increased computational cost is well justified by its substantially expanded capability. TAL$^+$ can predict parameters for models more than 4 times larger than previous methods, while also achieving significantly better initialization performance across most tasks.

**TAL presents superior parameter prediction ability across unseen tasks.** We evaluate TAL against LoGAH trained on ImageNet-1K and random initialization (RandInit) on a broader set of unseen datasets. Specifically, we use three datasets from distinct fields: Fashion MNIST (Xiao et al., 2017), a dataset of fashion item images; FER2013 (Goodfellow et al., 2013), a facial expression recognition dataset; and HAM10000m (Tschandl et al., 2018), a medical dataset for

Table 7: Performance of untrained models initialized with single-task tuning (TAL-st) and multi-task tuning (TAL) on Decathlon tasks.

| MODEL | METHOD | AIRC. | C100 | DPED | DTD | GSTR | OGLE | SVHN | UCF | FLWR |
|---|---|---|---|---|---|---|---|---|---|---|
| 12-TINY | TAL-ST | 1.95 | 5.99 | 90.00 | 6.65 | 82.08 | 0.03 | 27.22 | 13.78 | 10.20 |
| | TAL | **19.8** | **54.89** | **98.61** | **30.21** | **99.57** | **63.57** | **90.38** | **48.1** | **47.84** |
| 12-SMALL | TAL-ST | 2.79 | 8.66 | **89.56** | 8.67 | **99.29** | 0.09 | 21.84 | 21.84 | 7.84 |
| | TAL | **17.16** | **45.63** | 82.16 | **27.87** | 99.18 | **39.66** | **88.55** | **40.98** | **45.10** |

Table 8: Performance of untrained models on Decathlon datasets initialized with TAL without using task information(T.I.) in learngene or ancestry model(ans-net) on Decathlon datasets.

| METHOD | ANC-NET | T.I. | AVG ACC |
|---|---|---|---|
| TAL(W/O T.I.) | ✓ | ✗ | 40.71 |
| TAL(W/O ANS-NET) | ✗ | ✓ | 46.80 |
| TAL | ✓ | ✓ | 53.37 |

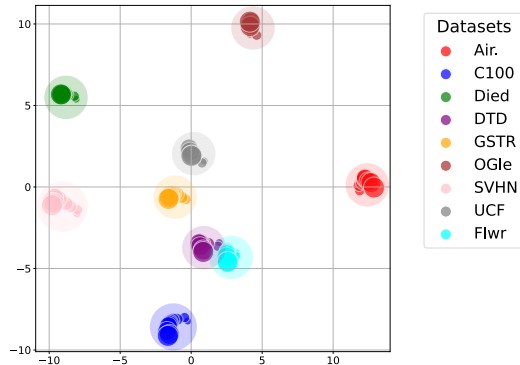

Figure 3: The computational graphs of all models generated by learngene on the Decathlon datasets. Each point represents a model computational graph. Different colors denote different tasks and the size of the point corresponds to the model's scale, with larger points indicating larger models.

final test accuracy. Therefore, in our analysis, we evaluate the performance of untrained models initialized with different methods on each dataset.

**The effect of multi-task training.** We investigate the impact of multi-task training by comparing TAL with TAL-st, where TAL-st sequentially fine-tunes the TAL model (pretrained on ImageNet) on each downstream task individually. We evaluate using two models: a 12-layer ViT-tiny (12-Tiny) and a 12-layer ViT-small (12-Small). As shown in Tab. 7, models initialized with TAL significantly outperform TAL-st across almost all Decathlon tasks.

**The effect of ancestry model and task information.** As shown in Tab.8, we conduct ablation studies by removing task information (T.I.) in learngene for TAL(w/o T.I.) and removing the guidance of ancestry model for TAL(w/o ans-net). Models initialized by TAL outperform TAL(w/o T.I.) and TAL(w/o ans-net) by **12.66%** and **6.57%** in test accuracy across Decathlon tasks, respectively. More comprehensive evaluation results are provided in Appendix .2

**Visualization of task-specific model computational graphs.** To verify the effectiveness of learngene in dynamically encoding the model computational graph under different task conditions, we visualize the output of learngene. We use 3-12 layers of ViT-small, a total of 10 test models and apply learngene to output their task-specific computational graphs on Decathlon Challenge datasets. We use the PCA (Abdi & Williams, 2010) method to map the high-dimensional features of the model computational graph to 2D space and visualize them. The result is shown in Fig.3. As the task information changes, the model's learned computational graph exhibits a significant clustering effect, the learned computational graphs of the model for different

tasks clearly cluster together in two-dimensional space. This indicates that learngene can effectively integrate task information while incorporating task information into the model computational graph.

## 6. Conclusion

In this paper, we propose a novel method called Task-Aware Learngene(TAL) that predicts model parameters conditioned on desired model scales and task-specific characteristics. Experimental results on various datasets demonstrated the effectiveness of TAL's ability to predict parameters. Untrained models initialized using TAL achieved significant improvements across various datasets compared to the previous GHN initialization methods. Remarkably, the accuracy of these untrained models even surpassed the performance of models trained using other initialization methods.

## Acknowledgements

This research was supported by the Jiangsu Science Foundation (BK20243012, BG2024036), the National Science Foundation of China (62125602, U24A20324, 92464301, 62406066), the Fundamental Research Funds

for the Central Universities (2242025K30024), Jiangsu Province Science Foundation for Youths (BK20241297), Taihu Lake Innovation Fund for the School of Future Technology of Southeast University, and the Big Data Computing Center of Southeast University.

## Impact Statement

This paper presents work whose goal is to advance the field of Machine Learning. There are many potential societal consequences of our work, none which we feel must be specifically highlighted here.

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

## .1. Appendix 1

The Visual Domain Decathlon Challenge tests the ability of visual recognition algorithms to handle images from different visual domains. It includes 10 datasets in total:

1. **ImageNet-1K** (IN-1K) is the largest dataset in the Decathlon Challenge, containing 1,000 categories and 1.2 million images.

2. **CIFAR-100** (C100) contains 60,000 $32 \times 32$ color images for 100 object categories.

3. **Aircraft** (Airc.) contains 100 images for each of 100 different aircraft model variants, such as the Boeing 737-400 and the Airbus A310.

4. **Daimler Pedestrian Classification** (DPed) consists of 50,000 grayscale pedestrian and non-pedestrian images, cropped and resized to $18 \times 36$ pixels.

5. **Describable Textures** (DTD) is a texture database consisting of 5,640 images, organized into 47 categories such as bubbly, cracked andmarbled.

6. **German Traffic Signs** (GTSRB) contains cropped images for 43 common traffic sign categories in different image resolutions.

7. **Omniglot** (OGlt) consists of 1,623 different handwritten characters from 50 unique alphabets.

8. **SVHN** is a real-world digit recognition dataset with around 70,000 $32 \times 32$ images.

9. **UCF101 Dynamic Images** (UCF) is an action recognition dataset of realistic human action videos, collected from YouTube. It contains 13,320 videos across 101 action categories. In the Decathlon Challenge, the videos are converted into images using Dynamic Image encoding, which summarizes each video into an image based on a ranking principle.

10. **Flowers102** (Flwr) is a fine-grained classification task with 102 flower categories from the UK, each consisting of 40 to 258 images.

The detailed statistics of the datasets can be found at `http://www.robots.ox.ac.uk/~vgg/decathlon/`.

## .2. Appendix 2

Here we present the detailed results of the analysis and ablation studies for TAL (w/o I.T.) and TAL (w/o ans-ant) methods. The initialization performance of TAL (w/o I.T.) and TAL (w/o ans-ant) methods on the Decathlon dataset is shown in Tab.9.

Table 9: Performance of untrained models on Decathlon datasets initialized with TAL without using task information(T.I.) in learngene or ancestry model(ans-net) on Decathlon datasets.

| MODEL | METHOD | AIRC. | C100 | DPED | DTD | GSTR | OGLE | SVHN | UCF | FLWR | AVG |
|---|---|---|---|---|---|---|---|---|---|---|---|
| 3-TINY | TAL(W/O ) | 4.05 | 17.61 | 72.65 | 11.12 | 65.9 | 1.76 | 35.77 | 2.25 | 9.61 | 24.52 |
| | TAL(W/O ANS-NET) | 3.39 | 49.92 | 92.74 | 20.80 | 99.39 | 0.18 | 90.2 | 11.37 | 50.69 | 46.52 |
| 6-TINY | TAL(W/O I.T.) | 12.18 | 40.62 | 72.89 | 23.30 | 99.20 | 52.57 | 88.77 | 2.87 | 34.22 | 47.40 |
| | TAL(W/O ANS-NET) | 3.27 | 49.86 | 92.38 | 20.05 | 99.35 | 0.17 | 90.34 | 12.09 | 51.18 | 46.52 |
| 12-TINY | TAL(W/O T.I.) | 10.23 | 36.09 | 65.83 | 17.13 | 96.00 | 31.28 | 83.92 | 3.02 | 25.10 | 40.96 |
| | TAL(W/O ANS-NET) | 2.88 | 49.45 | 93.01 | 21.22 | 99.31 | 0.15 | 90.43 | 12.19 | 51.08 | 46.64 |
| 3-SMALL | TAL(W/O T.I.) | 12.54 | 40.65 | 80.24 | 20.69 | 98.99 | 49.85 | 85.99 | 4.35 | 36.86 | 47.79 |
| | TAL(W/O ANS-NET) | 2.88 | 50.76 | 92.76 | 21.97 | 99.35 | 0.09 | 90.51 | 11.53 | 52.35 | 46.91 |
| 6-SMALL | TAL(W/O T.I.) | 13.68 | 42.43 | 77.19 | 22.61 | 99.30 | 52.65 | 89.28 | 3.28 | 37.75 | 48.69 |
| | TAL(W/O ANS-NET) | 3.12 | 50.91 | 93.10 | 22.29 | 99.29 | 0.14 | 90.57 | 12.81 | 52.25 | 47.16 |
| 12-SMALL | TAL(W/O T.I.) | 11.73 | 37.67 | 60.17 | 16.97 | 55.30 | 23.35 | 76.45 | 2.97 | 29.22 | 34.87 |
| | TAL(W/O ANS-NET) | 2.88 | 50.86 | 93.52 | 21.76 | 99.34 | 0.12 | 90.64 | 12.19 | 52.16 | 47.05 |

## .3. Appendix 3

We discuss a simplified case of our TAL method and provide a theoretical derivation to complete this missing part.

## 1. Problem Definition and Optimization Objective

We define the following setup:

- **Hypernetwork** $H : \Theta \to \mathbb{R}^d$ A multilayer perceptron (MLP) that maps from parameter space $\Theta$ to model parameter space $\mathbb{R}^d$, generating parameters $p = H(\theta)$.

- **Model** $M$ Also an MLP, using parameters $p$ to perform a binary $(0,1)$ classification task and compute the loss $\mathcal{L}(p)$.

The optimization objective is to train the hypernetwork $H$ to minimize the cross-entropy loss:

$$\min_{\theta} \mathcal{L}(H(\theta)) = \mathbb{E}_{(x,y) \sim \mathcal{D}}[-y\log(f_M(x;H(\theta))) - (1-y)\log(1 - f_M(x;H(\theta)))]$$

## 2. Convergence Analysis

**Theorem 1 (Convergence to Stationary Point):** Assume the following conditions hold:

- The loss function $\mathcal{L}(p)$ is $\beta$-smooth

- Hypernetwork $H(\theta)$ is $L_H$-Lipschitz continuous

- The composed function $\mathcal{L}(H(\theta))$ has bounded gradients

Then, using gradient descent with learning rate $\eta < \frac{2}{L_H \beta}$, after $T$ iterations:

$$\min_{t=0,1,...,T-1} \|\nabla_\theta \mathcal{L}(H(\theta_t))\|^2 \le \frac{2(\mathcal{L}(H(\theta_0)) - \mathcal{L}(H(\theta^*)))}{T\eta}$$

**Proof:**
By $\beta$-smoothness of $\mathcal{L}$ and $L_H$-Lipschitz continuity of $H$, the composite function $\mathcal{L}(H(\theta))$ is $(L_H \beta)$-smooth. For a $(L_H \beta)$-smooth function, when using gradient descent with learning rate $\eta < \frac{2}{L_H \beta}$:

$$\mathcal{L}(H(\theta_t)) - \mathcal{L}(H(\theta_{t+1})) \ge \eta \left(1 - \frac{L_H \beta \eta}{2}\right) \|\nabla_\theta \mathcal{L}(H(\theta_t))\|^2$$

Summing over $t = 0,1,...,T-1$ and rearranging:

$$\sum_{t=0}^{T-1} \|\nabla_\theta \mathcal{L}(H(\theta_t))\|^2 \le \frac{\mathcal{L}(H(\theta_0)) - \mathcal{L}(H(\theta_T))}{\eta \left(1 - \frac{L_H \beta \eta}{2}\right)}$$

$$\le \frac{\mathcal{L}(H(\theta_0)) - \mathcal{L}(H(\theta^*))}{\eta \left(1 - \frac{L_H \beta \eta}{2}\right)}$$

Since $\eta < \frac{2}{L_H \beta}$ implies $1 - \frac{L_H \beta \eta}{2} > 0$, and using the minimum gradient norm:

$$T \cdot \min_{t=0,1,...,T-1} \|\nabla_\theta \mathcal{L}(H(\theta_t))|^2 \le \sum_{t=0}^{T-1} \|\nabla_\theta \mathcal{L}(H(\theta_t))\|^2$$

$$\le \frac{\mathcal{L}(H(\theta_0)) - \mathcal{L}(H(\theta^*))}{\eta \left(1 - \frac{L_H \beta \eta}{2}\right)}$$

With proper learning rate, $1 - \frac{L_H \beta \eta}{2} \geq \frac{1}{2}$, resulting in:

$$\min_{t=0,1,\ldots,T-1} \|\nabla_\theta \mathcal{L}(H(\theta_t))\|^2 \leq \frac{2(\mathcal{L}(H(\theta_0)) - \mathcal{L}(H(\theta^*)))}{T\eta}$$

This shows that as $T \to \infty$, the gradient norm approaches zero, indicating convergence to a stationary point. $\square$

**Corollary 1 (Convergence Rate):** Under the conditions of Theorem 1, the gradient descent method converges to a stationary point at a rate of $\mathcal{O}(1/\sqrt{T})$.

### 3. Optimality Analysis

**Theorem 2 (Universal Approximation):** If the hypernetwork $H$ is a sufficiently wide and deep MLP, then for any $\delta > 0$ and any target parameter $p^* \in \mathbb{R}^d$, there exists a parameter $\theta$ such that:

$$\|H(\theta) - p^*\| < \delta$$

**Proof:**
According to the universal approximation theorem, a sufficiently wide MLP can approximate any continuous function on a compact domain to arbitrary precision. Treating the mapping from a fixed input to the target parameter vector $p^*$ as a constant function, there exists an MLP architecture for $H$ and parameters $\theta$ such that $\|H(\theta) - p^*\| < \delta$ for any $\delta > 0$. $\square$

**Corollary 2 (Approximation of Optimal Loss):** Under the conditions of Theorem 3, for any $\epsilon > 0$, there exists a hypernetwork $H$ and parameters $\theta$ such that:

$$\mathcal{L}(H(\theta)) - \mathcal{L}(p^*) < \epsilon$$

**Conclusion**

Our analysis of hypernetwork optimization for binary classification with MLPs has established:

- **Convergence:** Gradient-based optimization of hypernetworks converges to stationary points at a rate of $\mathcal{O}(1/\sqrt{T})$ under standard smoothness assumptions.

- **Approximation Capability:** Sufficiently expressive hypernetworks can approximate optimal model parameters to arbitrary precision.

