# OpenReview forum: "Learngene Tells You How to Customize: Task-Aware Parameter Initialization at Flexible Scales"
_ICML.cc/2025/Conference — ICML 2025 poster_

### Official Review · Reviewer_56qr · 2025-03-11

**Overall Recommendation:** 3

**Summary:**

This paper proposes a novel parameter initialization method called TAL, aiming to enhance the initialization effect of models for different tasks. Building upon the previous GHN and Learngene frameworks, TAL addresses their limitations. Although the GHN method is effective, it performs inadequately when dealing with large-scale models and requires retraining for each new task. In contrast, by integrating the Learngene framework with a task-aware mechanism, TAL is able to share knowledge across multiple tasks, thereby improving the accuracy of parameter initialization and eliminating the need for separate training for each task. The experimental results demonstrate that TAL outperforms methods such as GHN and LoGAH in both visual and natural language processing tasks.

**Claims And Evidence:**

The argument of this paper is that the latest graph hypernetwork methods, such as LoGAH, exhibit poor initialization accuracy on IN-1K and CIFAR-100 when initializing larger models. Furthermore, the author presents the accuracy on IN-1K and other datasets in the experimental section, reasonably addressing the proposed issues.

**Essential References Not Discussed:**

N/A

**Experimental Designs Or Analyses:**

The experimental design is relatively reasonable, covering not only diverse visual tasks but also language tasks, which demonstrates the certain universality of the method proposed in this paper.

**Methods And Evaluation Criteria:**

The proposed methods and evaluation criteria are reasonable for the current problem.

**Other Comments Or Suggestions:**

1 In Formula 6, Laux needs to be further explained.

2 It is recommended that the author optimize the logical structure of the text and provide a clearer elaboration of the two core concepts, learngene and computational graph.

**Other Strengths And Weaknesses:**

The experiments are sufficient, but the analysis of the experiments is rather hasty. For example, why does TAL+ perform worse than TAL in some cases (the second row of 12-Tiny in Table 1)?

**Questions For Authors:**

1 What is the difference between the Decoder and Encoder in Figure 2?

2 Why does TAL+ perform worse than TAL in some cases, such as the second row of 12-Tiny and the second row of 12-Small in Table 1? Additionally, why does TAL+ far outperform the other three methods in Table 1? And why does TAL+ perform much worse than TAL on AIRC. and OGLE in Table 2?

3 How much additional overhead does the two-stage training (Figure 2ab) incur compared to other comparative methods?

4 How is an entire ViT initialized? Is it initialized layer by layer or are the parameters of the entire network initialized directly?

**Relation To Broader Scientific Literature:**

Previous methods, such as LoGAH, show poor initialization accuracy on IN-1K and CIFAR-100 when initializing larger models. The TAL proposed by the author performs better on datasets like IN-1K.

**Theoretical Claims:**

The theoretical basis is the learngene. Additionally, the concept of the computational graph is also mentioned. It is necessary for the author to elaborate on these concepts in more detail.

---

> ### Author Rebuttal · Authors · 2025-03-31
>
> We thank you for your reviews and address your concerns as follows.
> ### Q1
> The author should elaborate on the learngene theory and the computational graph in more detail.
> ### A1
> We elaborate on the concept of Learngene in the introduction and related work of the paper. The essence of the Learngene framework lies in condensing critical knowledge from an ancestry model to initialize downstream models. Our implementation uses an encoder-decoder architecture to inherit knowledge from the ancestry model and generate downstream models.
> Regarding computational graphs, we provide a detailed description in lines 155-164 of the paper and add relevant examples in the appendix to enhance the reader's understanding of the concept. The computation graphs primarily include node\_feat, node\_info, and edges\_embedding. To clarify, we provide an example using the ViT model to illustrate how computation graphs are constructed.
>
> ```python
> node\_feat:
> tensor([
>     [9],   # 'input'
>     [13],  # 'pos\_enc'
>     [5],   # 'msa'
>     [12],  # 'ln'
>     [3],   # 'linear'
>     [7],   # 'sum'
>     [3],   # 'linear'
> ])
> node\_info:
> [
>     [[0, 'input', 'input', None, False, False]],
>     [[1, 'pos\_enc', 'pos\_enc', (1, 768, 14, 14), False, False]],
>     [[2, 'msa', 'msa', (1, 768, 14, 14), False, False]],
>     [[3, 'ln', 'ln', (1, 768), False, False]],
>     [[4, 'linear', 'linear', (768, 3072), False, False]],
>     [[5, 'sum', 'sum', None, False, False]],
>     [[6, 'linear', 'linear', (3072, 768), False, False]],
> ]
> edges embedding:
> tensor([
>     [0, 1, 1],  # input -> pos\_enc
>     [1, 2, 1],  # pos\_enc -> msa
>     [2, 3, 1],  # msa -> ln
>     [3, 4, 1],  # ln -> linear
>     [4, 5, 1],  # linear -> sum
>     [5, 6, 1],  # sum -> linear
> ])
> ```
> Hopefully, this simple example helps clarify the concept.
> ### Q2
> Why does TAL+ perform worse than TAL in some cases.
> ### A2
> The primary distinction between TAL and TAL+ lies in the size of the model library used during training. TAL+ leverages a larger model library, which generally improves performance, as evidenced by its superior results in 7 out of 9 tasks in Decathlon datasets. However, in certain cases, such as the second row of 12-Tiny and 12-Small in Table 1, TAL+ underperforms TAL. This discrepancy arises from differences in sample distribution—since TAL+ introduces more large models into training, the proportion of Tiny-sized samples decreases significantly. This imbalance affects TAL+’s ability to adapt to smaller models.
>
> In Table 2, TAL+ performs worse than TAL on Airc and OGle. Both datasets have high class diversity but limited samples per class, with OGle being a few-shot task comprising 1623 unique characters from 50 alphabets. The key factor here is model capacity—TAL+ models trained on a larger model library tend to have more parameters, which can be less effective in scenarios with limited training samples. The increased capacity of TAL+ models may hinder generalization on these high-diversity, low-resource classification tasks. Despite these limitations, TAL+ is capable of handling larger models and achieves superior performance on most tasks.
> ### Q3
> In Formula 6, Laux needs to be further explained.
> ### A3
> The specific expression and calculation of Laux in Formula 6 is described in detail in Formula 4. Briefly, we use the output of the ancestry model as a soft label to align the model's output initalized by TAL.
> ### Q4
> What is the difference between the Decoder and Encoder in Figure 2?
> ### A4
> We introduce Encoder and Decoder in section4 of the paper. The Encoder learns and characterizes the model's computational graph, capturing its structure and parameter relationships. It adopts a Transformer architecture to efficiently handle long-range dependencies in graph structures.
>
> The Decoder decodes the learned graph representations to generate the parameter matrices for target models. These matrices are then trimmed or copied to match the required model shape. The Decoder uses MLPs to convert graph representations into parameter values.
> ### Q5
> How much additional overhead does the two-stage training incur compared to other methods?
> ### A5
> Table 4 shows the computational overhead across methods, with TAL's total time of 36.19 hours including both training stages (28.47 hours for stage 1 and 7.72 hours for stage 2). TAL requires fewer training epochs in stage 2 than comparison methods, resulting in less total training time.
> ### Q6
> How is an entire ViT initialized?
> ### A6
> Our approach initializes the ViT model layer by layer. During the TAL forward pass, the encoder processes the computational graph to learn node embeddings, then the decoder generates parameters through internal loops where the decoder is called multiple times for different parameter groups of different layers (attention weights, MLP weights, etc.). Despite being sequential, this initialization process completes in under 0.1 seconds for an entire ViT model, preserving architectural information while maintaining efficiency.

---

> > ### Comment · Reviewer_56qr · 2025-04-03
> >
> > Thank you for the author's reply. However, the author has not addressed my core concerns.
> >
> > For example, in A2, the author did not provide a detailed response regarding the discrepancies on the Airc dataset. Additionally, when comparing Tables 2 and 3, taking the 12-small, C100, and SVHN datasets as examples (the number of samples in the two datasets is similar), before training, TAL+ performed better than TAL on C100 and SVHN. But after training, TAL did not surpass TAL+ on C100 while it did surpass TAL+ on SVHN. Why is this the case?
> >
> > I hope the author can conduct further analysis and provide explanations for the results in the experiments.

---

> > > ### Author Response · Authors · 2025-04-07
> > >
> > > We thank you for your reviews and address your concerns as follows.
> > >
> > > Aircraft (Airc.) contains 100 images for each of 100 different aircraft model variants, such as the Boeing 737-400 and the Airbus A310. This dataset's limited size makes it insufficient for fully leveraging the capabilities of ViT models, which typically require larger training sets.
> > >
> > > For further analysis of CIFAR100 and SVHN results:
> > > # Performance Comparison
> > > We set up different seeds and employed the TAL and TAL+ methods to initialize ViT-tiny and ViT-small models for the CIFAR100 and SVHN datasets, respectively. After initialization, all models were trained for 100 epochs. The results from multiple rounds of testing are presented below.
> > >
> > > **Table 1: Performance comparison of TAL+ and TAL methods across three rounds for ViT-tiny and ViT-small models on SVHN.**
> > >
> > > | Method | Round 1 | Round 2 | Round 3 |
> > > |--------|---------|---------|---------|
> > > | **ViT-tiny** |
> > > | TAL+   | 91.07   | 90.86   | 90.94   |
> > > | TAL    | 90.76   | 91.03   | 90.31   |
> > > | **ViT-small** |
> > > | TAL+   | 91.97   | 91.57   | 91.72   |
> > > | TAL    | 91.64   | 91.95   | 92.02   |
> > >
> > > **Table 2: Performance comparison of TAL+ and TAL methods across three rounds for ViT-tiny and ViT-small models on Cifar100.**
> > >
> > > | Method | Round 1 | Round 2 | Round 3 |
> > > |--------|---------|---------|---------|
> > > | **ViT-tiny** |
> > > | TAL+   | 57.90   | 58.78   | 58.90   |
> > > | TAL    | 57.71   | 57.98   | 59.80   |
> > > | **ViT-small** |
> > > | TAL+   | 60.51   | 60.24   | 60.77   |
> > > | TAL    | 61.40   | 61.63   | 60.49   |
> > >
> > > To assess statistical significance, we analyzed results from multiple rounds under different random seeds. We tested whether models initialized with TAL and TAL+ would show significant differences in post-training performance, computing p-values for each scenario.
> > >
> > > **Table 3: Statistical significance (p-values) comparing TAL+ and TAL methods on CIFAR-100 and SVHN.**
> > >
> > > | Model     | CIFAR-100 | SVHN    |
> > > |-----------|-----------|---------|
> > > | ViT-tiny  | 0.9674    | 0.3846  |
> > > | ViT-small | 0.3103    | 0.6551  |
> > >
> > > In statistical analysis, a p-value greater than 0.05 typically indicates that we cannot reject the null hypothesis. The table above shows that the p-values in all cases are substantially greater than 0.05. Based on these paired t-test results, we conclude that there is no statistically significant difference in performance between the TAL+ and TAL methods after training, suggesting that both initialization approaches ultimately produce comparable results.
> > >
> > > # Parameter Characteristics Analysis
> > >
> > > Although there is no significant difference in post-training performance between TAL and TAL+ initialization methods, models initialized with TAL+ perform significantly better than those initialized with TAL when evaluated in their untrained state. To further investigate this phenomenon, we analyzed the parameters of models initialized by both methods before and after training on the CIFAR100 dataset.
> > >
> > > We examined the following metrics[1], [2]:
> > >
> > > - **Parameter sparsity:** the proportion of parameters close to zero in the model
> > > - **Parameter diversity:** cosine distance between models initialized with different random seeds
> > > - **Parameter change magnitude:** the relative degree of change in parameters before and after training
> > >
> > > The analysis results are presented below:
> > >
> > > **Table 4: Parameter characteristics comparison between TAL and TAL+ methods for ViT-Small and ViT-Tiny models on CIFAR100.**
> > >
> > > | Metrics | ViT-Small |        | ViT-Tiny |        |
> > > |---------|-----------|--------|----------|--------|
> > > |         | TAL       | TAL+   | TAL      | TAL+   |
> > > | Initial Parameter Sparsity | 79.97% | 77.99% | 78.89% | 76.00% |
> > > | Post-training Parameter Sparsity | 79.48% | 77.50% | 78.10% | 75.17% |
> > > | Parameter Change Magnitude | 1.69 | 2.73 | 2.45 | 2.64 |
> > > | Post-training Parameter Diversity | 0.0485 | 0.0896 | 0.0424 | 0.0789 |
> > >
> > > Despite TAL+ demonstrating superior performance in the untrained state, our experimental results indicate that both methods yield comparable performance after training. This phenomenon may be attributed to TAL+ converging too early to a specific region of the solution space, thereby limiting exploration of potentially better solutions. In contrast, the higher stochasticity of TAL enables the model to explore a broader range of the parameter space. Additionally, the higher parameter sparsity observed in TAL may provide an implicit regularization effect that helps the model achieve cleaner and better-generalized solutions, which explains why its final performance matches or occasionally exceeds that of TAL+.
> > >
> > >
> > > [1]:Knyazev, Boris, et al. "Parameter prediction for unseen deep architectures." Advances in Neural Information Processing Systems 34 (2021): 29433-29448.
> > >
> > > [2]:Knyazev, Boris, Doha Hwang, and Simon Lacoste-Julien. "Can we scale transformers to predict parameters of diverse imagenet models?." International Conference on Machine Learning. PMLR, 2023.

---

### Official Review · Reviewer_AdW5 · 2025-03-13

**Overall Recommendation:** 3

**Summary:**

This paper addresses the high computational and storage overheads involved in training large pretrained models by focusing on effective parameter initialization. Building on recent advances in Graph HyperNetworks (GHN) and the Learngene framework, the authors propose a novel method called Task-Aware Learngene (TAL). TAL is designed to capture shareable, task-specific knowledge from a well-trained “ancestry” model and use it to predict initial parameters for models of varying scales.

**Claims And Evidence:**

Yes

**Essential References Not Discussed:**

An important Task-Aware Parameter  initialization works ins not discussed:  Learning to Generate Parameters of ConvNets for Unseen Image Data (TIP 2024)

**Experimental Designs Or Analyses:**

Experiments are conducted on multiple vision and language tasks with a variety of model scales (e.g., ViT-Tiny, ViT-Small, ViT-Base, and GPT-2 models with different depths).
However, an important Task-Aware Parameter initialization baseline is not compared:  Learning to Generate Parameters of ConvNets for Unseen Image Data (TIP 2024)

**Methods And Evaluation Criteria:**

The proposed TAL method leverages an encoder–decoder architecture where the encoder (learngene) extracts and transfers task-specific knowledge from an ancestry model, and a task hypernetwork generates task-specific bias parameters for a task-specific layer.
The method is evaluated on both vision and language tasks using standard model architecture datasets (e.g., ViTs-1K, ViTs+-1K, GPTS-1K) and downstream benchmarks (ImageNet-1K, Decathlon, MRPC, COLA, RTE, IMDB).

**Other Comments Or Suggestions:**

N.A

**Other Strengths And Weaknesses:**

The proposed method is sound and interesting.
However, there are several weaknesses:

An important and recent Task-Aware Parameter initialization works [1] is not compared.

There is limited discussion on the scalability of TAL when applied to extremely large models, which could be crucial for real-world applications.

The models used in this paper is too small and shallow. Deeper network like ResNet-34 is encourgaed to be used for testing, as in previous works [1].

The theoretical underpinning of the method, including convergence properties and optimality guarantees, is not fully explored.

[1]   Learning to Generate Parameters of ConvNets for Unseen Image Data (TIP 2024)

**Questions For Authors:**

N/A

**Relation To Broader Scientific Literature:**

AL builds on prior work in hypernetworks and Task-Aware Parameter  initialization methods.

**Theoretical Claims:**

N/A

---

> ### Author Rebuttal · Authors · 2025-03-31
>
> We thank you for your reviews and address your concerns as follows.
> ### Q1
> An important Task-Aware Parameter initialization baseline is not compared: Learning...(TIP 2024). The models used in this paper is too small and shallow.
> ### A1
> Thank you for pointing this out. We will cite it in the relevant section. As for the model size predicted by TAL+, our framework has been tested on 12-layer ViT-Base models with 86M parameters, far exceeding ResNet-34’s 22M. It also handles deeper architectures, including the 16-layer ViT-Base (115M), which achieves 31.52% accuracy on ImageNet-1K without further training. Scaling up to an 18-layer ViT-Base (129M), the model maintains a competitive 22.25% accuracy. These experiments validate TAL’s effectiveness on larger models.
> ### Q2
> There is limited discussion on the scalability of TAL.
> ### A2
> Our experiments cover various model sizes, including ViT-Base (12 layers). Indeed, TAL is designed for scalability and can support larger models. We are currently testing it on large language models (e.g., LLaMA) and will present the results in future work.
> ### Q3
> The theoretical underpinning of the method is not fully explored.
> ### A3
> We discuss a simplified case of our TAL method and provide a theoretical derivation. We reach the following conclusions:
>
> - **Convergence:** Gradient-based optimization of hypernetworks converges to stationary points under standard smoothness assumptions.
>
> - **Approximation Capability:** Sufficiently expressive hypernetworks can approximate optimal model parameters to arbitrary precision.
>
> **1. Problem Definition and Optimization Objective**
> We define the following setup:
> - **Hypernetwork** $H: \Theta \rightarrow \mathbb{R}^d$  A multilayer perceptron (MLP) that maps from parameter space $\Theta$ to model parameter space $\mathbb{R}^d$, generating parameters $p = H(\theta)$.
> - **Model** $M$  Also an MLP, using parameters $p$ to perform a binary (0,1) classification task and compute the loss $\mathcal{L}(p)$.
> The optimization objective is to train the hypernetwork $H$ to minimize the cross-entropy loss:
> $$\min_{\theta} \mathcal{L}(H(\theta)) = \mathbb{E}_{(x,y) \sim \mathcal{D}} \left[ -y\log(f_M(x; H(\theta))) - (1-y)\log(1-f_M(x; H(\theta))) \right]$$
>
> **2. Convergence Analysis**
>
> **Theorem 1 (Convergence to Stationary Point):** Assume the following conditions hold:
> - The loss function $\mathcal{L}(p)$ is $\beta$-smooth
> - Hypernetwork $H(\theta)$ is $L_H$-Lipschitz continuous
> - The composed function $\mathcal{L}(H(\theta))$ has bounded gradients
> Then, using gradient descent with learning rate $\eta < \frac{2}{L_H\beta}$, after $T$ iterations:
> $$\min_{t=0,1,...,T-1} \||\nabla_{\theta} \mathcal{L}(H(\theta_t))\||^2 \leq \frac{2(\mathcal{L}(H(\theta_0)) - \mathcal{L}(H(\theta^*)))}{T\eta}$$
>
> **Proof:**
> By $\beta$-smoothness of $\mathcal{L}$ and $L_H$-Lipschitz continuity of $H$, the composite function $\mathcal{L}(H(\theta))$ is $(L_H\beta)$-smooth. For a $(L_H\beta)$-smooth function, when using gradient descent with learning rate $\eta < \frac{2}{L_H\beta}$:
> $$\mathcal{L}(H(\theta_t)) - \mathcal{L}(H(\theta_{t+1})) \geq \eta\left(1 - \frac{L_H\beta\eta}{2}\right)\||\nabla_{\theta}\mathcal{L}(H(\theta_t))\||^2$$
> Summing over $t=0,1,...,T-1$ and rearranging:
> $$\sum_{t=0}^{T-1}\||\nabla_{\theta}\mathcal{L}(H(\theta_t))\||^2 \leq \frac{\mathcal{L}(H(\theta_0)) - \mathcal{L}(H(\theta_T))}{\eta\left(1 - \frac{L_H\beta\eta}{2}\right)}$$
> $$\leq \frac{\mathcal{L}(H(\theta_0)) - \mathcal{L}(H(\theta^*))}{\eta\left(1 - \frac{L_H\beta\eta}{2}\right)}$$
> Since $\eta < \frac{2}{L_H\beta}$ implies $1 - \frac{L_H\beta\eta}{2} > 0$, and using the minimum gradient norm:
> $$T \cdot \min_{t=0,1,...,T-1} \||\nabla_{\theta}\mathcal{L}(H(\theta_t))\||^2 \leq \sum_{t=0}^{T-1}\||\nabla_{\theta}\mathcal{L}(H(\theta_t))\||^2$$
> $$\leq \frac{\mathcal{L}(H(\theta_0)) - \mathcal{L}(H(\theta^*))}{\eta\left(1 - \frac{L_H\beta\eta}{2}\right)}$$
> With proper learning rate, $1 - \frac{L_H\beta\eta}{2} \geq \frac{1}{2}$, resulting in:
> $$\min_{t=0,1,...,T-1} \||\nabla_{\theta} \mathcal{L}(H(\theta_t))\||^2 \leq \frac{2(\mathcal{L}(H(\theta_0)) - \mathcal{L}(H(\theta^*)))}{T\eta}$$
> This shows that as $T \to \infty$, the gradient norm approaches zero, indicating convergence to a stationary point. $\square$
>
> **3. Optimality Analysis**
>
> **Theorem 2 (Universal Approximation):** If the hypernetwork $H$ is a sufficiently wide and deep MLP, then for any $\delta > 0$ and any target parameter $p^* \in \mathbb{R}^d$, there exists a parameter $\theta$ such that:
> $$\||H(\theta) - p^*\|| < \delta$$
>
> **Proof:**
> According to the universal approximation theorem, a sufficiently wide MLP can approximate any continuous function on a compact domain to arbitrary precision. Treating the mapping from a fixed input to the target parameter vector $p^*$ as a constant function, there exists an MLP architecture for $H$ and parameters $\theta$ such that $\||H(\theta) - p^*\|| < \delta$ for any $\delta > 0$. $\square$

---

> > ### Comment · Reviewer_AdW5 · 2025-04-03
> >
> > Thanks for the response. My concerns have been addressed.

---

### Official Review · Reviewer_7VCD · 2025-03-15

**Overall Recommendation:** 3

**Summary:**

Authors propose TAL, an encoder-decoder method to generate parameters for initializing models of various sizes given a model architecture and a task embedding.

**Claims And Evidence:**

1. Yes

**Essential References Not Discussed:**

n/a

**Experimental Designs Or Analyses:**

1. Experiments seem to be reasonable and demonstrating TAL's effectiveness over GHN-based method.

**Methods And Evaluation Criteria:**

1. The method relies on a single ancestry model, which might be problematic as the latest model might serve as a better ancestry model, and then requires retraining. And since the entire framework requires a certain level of "pretraining" before it can serve as an initialization generator, this inherently brings a conflict. It is hard to achieve "train once and use forever". Maybe authors can show that when serving as initialization, the choice of ancestry model isn't that important.
2. Authors seem to include ViTs of various depths in their experiments as a demonstration for need of models of different sizes. What would be a potential use case for models with various depths? (i.e. wider models or deeper models) As far as I am concerned, a certain depth-to-width ratio is usually adopted and people rarely change that during application.

**Other Comments Or Suggestions:**

n/a

**Other Strengths And Weaknesses:**

n/a

**Questions For Authors:**

n/a

**Relation To Broader Scientific Literature:**

n/a

**Theoretical Claims:**

n/a

---

> ### Author Rebuttal · Authors · 2025-03-31
>
> We thank you for your reviews and address your concerns as follows.
> ### Q1
>
> The method relies on a single ancestry model, which might be problematic as the latest model might serve as a better ancestry model, and then requires retraining. And since the entire framework requires a certain level of "pre-training" before it can serve as an initialization generator, this inherently brings a conflict. It is hard to achieve "train once and use forever". Maybe authors can show that when serving as initialization, the choice of ancestry model isn't that important.
>
> ### A1
>
> Our approach allows flexibility in choosing the ancestry model based on the TAL training datasets, including the consideration of using the most recent model. It should be clarified that our approach does not involve any retraining of the ancestry model, instead we retrain the TAL encoder-decoder framework, not the ancestry model itself.
> ### Q2
>
> Authors seem to include ViTs of various depths in their experiments as a demonstration for need of models of different sizes. What would be a potential use case for models with various depths? (i.e. wider models or deeper models) As far as I am concerned, a certain depth-to-width ratio is usually adopted and people rarely change that during application.
>
> ### A2
>
> Our approach employs a structured scaling strategy rather than a fixed depth-to-width ratio. We sample layers from 3-12 and hidden dimensions with a step size of 32, ranging from 192-768 across all model depths. Attention heads are selected to ensure divisibility with hidden dimensions, maintaining architectural integrity. This parameterization offers greater freedom in depth and width combinations while still preserving architectural coherence, enabling the generation of models with diverse parameter counts tailored to various computational budgets. This flexibility addresses deployment needs across resource-constrained devices to high-performance environments.

---

### Official Review · Reviewer_FFPW · 2025-03-15

**Overall Recommendation:** 2

**Summary:**

This paper presents Task-Aware Learngene (TAL), designed to initialize large models via parameter prediction. To accomplish this, the authors first employ an encoder-decoder architecture for the TAL model and train it under the supervision of an ancestry model to facilitate knowledge transfer. Subsequently, with the aim of improving the multi-task generalization ability of downstream models, the authors fine-tune the trained TAL model on a multi-task dataset, thereby acquiring multi-task knowledge from the ancestry model. Comparative evaluations against alternative methods (such as GHN-3 and LoGAH) underscore the efficacy of TAL across various scenarios.

**Claims And Evidence:**

In the right column of lines 36–40, the authors state that “…its effectiveness diminishes when initializing larger models like ViT-base,” a claim supported by Table 1. However, as indicated in the same table, the proposed TAL method also fails to initialize the ViT-base model on untrained configurations, achieving a very low-performance value of 0.1, identical to GHN-3 and LoGAH. While the TAL+ model outperforms other methods, its improvement appears to stem from training on an enhanced dataset (lines 236–244). My concern centers on whether all methods in Table 1 were trained on the same dataset. If so, why does only the TAL+ method perform well, while the performance of other methods is nearly random? If not, this comparison is problematic, as it evaluates methods trained on different datasets, thereby undermining the validity of the claim.

**Essential References Not Discussed:**

None.

**Experimental Designs Or Analyses:**

I have reviewed the experimental design and analyses outlined in Section 5, and the following concerns arise: \
1.	Lack of Supporting Evidence for Convergence Claims: The authors assert that “Models initialized with TAL/TAL+ converge faster and outperform …” (line 327). However, this claim is not substantiated by detailed experimental results, figures, or tables. While higher accuracy values are presented, no evidence is provided to demonstrate faster convergence, which is a critical aspect of the claim. \
2.	Limited Validation of Untrained Initialization Effectiveness: The effectiveness of TAL/TAL+ in initializing models without training has only been validated on the Decathlon tasks. This narrow scope raises questions about the method’s broader applicability, as it has not been tested on other NLP tasks or diverse domains. Such validation is essential to establish the generalizability of the approach. \
3.	Incomplete Comparison with Task Information: Task information appears to be a feature that could also benefit other methods, such as GHN-3. However, the experiments in Table 8 focus solely on the proposed TAL method without comparing it to other methods that incorporate task information. This omission limits the ability to assess whether the observed improvements are unique to TAL or could be achieved by enhancing existing methods with similar information.

**Methods And Evaluation Criteria:**

Yes.

**Other Comments Or Suggestions:**

None.

**Other Strengths And Weaknesses:**

Strengths: \
1.	The proposed TAL effectively integrates multi-task knowledge from an ancestry model via an encoder-decoder structure and multi-task fine-tuning, facilitating robust parameter prediction for diverse downstream tasks. \
2.	Empirical comparisons with methods like GHN-3 and LoGAH suggest TAL’s consistent performance improvements and highlight its potential for broader applications.

Weaknesses: \
In addition to the experimental issues noted above, I have a few further concerns: \
1.	Insufficient Motivation for the Ancestry Model: The rationale for introducing the ancestry model has not been fully articulated. A more detailed justification of its role and necessity would strengthen the paper’s foundation. \
2.	Ambiguity in the Concept of “Learngene”: The term “Learngene” is somewhat confusing in this context. In prior literature, it appears to refer to “critical components from the ancestry model used to initialize models of various sizes.” However, in this paper, it seems to function more as a “critical component for a parameter prediction model, which is learned through multi-task tuning and conditioned on task embeddings.” This shift in interpretation raises questions about whether the observed performance improvements are truly attributable to the “Learngene” concept or are instead driven by the multi-task tuning process and the incorporation of task embeddings.

**Questions For Authors:**

None.

**Relation To Broader Scientific Literature:**

Task-Aware Learngene (TAL) builds upon and extends existing research in model initialization, knowledge distillation, and multi-task learning. Several prior works, such as GHN-3 and LoGAH, have explored ways to generate or predict parameters for a great diversity of models. TAL differs with these earlier studies by leveraging an encoder-decoder structure and an ancestry model for knowledge transfer, as well as its explicit use of multi-task fine-tuning to enhance downstream generalization. By effectively integrating knowledge from an ancestry model and refining it through multi-task datasets, TAL demonstrates that parameter prediction for large models can achieve strong performance across diverse situations.

**Theoretical Claims:**

The paper does not contain any formal proofs or theorems.

---

> ### Author Rebuttal · Authors · 2025-03-31
>
> We thank you for your reviews and address your concerns as follows.
> ### Q1
> Were all methods trained on the same dataset? If so, why does only TAL+ perform well? If not, the comparison is unfair, undermining the claim's validity.
> ### A1
> TAL and GHN-3, LoGAH do use the same model training dataset, ViTs-1K, which ensures a fair comparison between them.
> The results in Table 1 and Table 2 of the paper clearly demonstrate the significant advantage of TAL under this fair comparison, as TAL outperforms the comparison method GHN-3/LoGAH on almost all datasets for the task of image classification.
> Beyond this, TAL+ introduces the augmented dataset ViTs+-1K to further enhance large model initialization, making the use of an augmented dataset one of our key contributions.
> ### Q2
> Lack of Supporting Evidence for Convergence Claims.
> ### A2
> We use 12-layer ViT-Small initialized with five different methods and train it on Decathlon datasets. We plot the training loss versus epochs to observe convergence speed. The table below presents results for the UCF dataset, where we observe that ViT models initialized with TAL/TAL+ methods demonstrate significantly faster convergence. Similar trends are observed across other datasets. The corresponding convergence plots for all datasets will be included in the appendix, as images cannot be displayed in the response.
> |**Epoch**|**10**|**20**|**30**|**40**|**50**|**60**|**70**|**80**|**90**|**100**|
> |---|---|---|---|---|---|---|---|---|---|---|
> |RandInit|59.9801|52.7643|45.5558|33.7087|17.9540|6.8132|2.9012|1.9813|1.6555|1.0367|
> |GHN-3|123.9892|114.4703|106.5648|96.6006|87.0042|75.6672|60.6773|47.0442|33.3951|23.7440|
> |LoGAH|116.4734|109.3229|101.4643|94.3091|86.6136|76.6659|65.2919|50.5523|37.1268|23.7927|
> |TAL|8.4604|7.0408|3.1283|2.2561|1.8384|1.5562|1.0961|0.8141|0.9439|0.7467|
> |TAL+|2.0044|2.7601|0.9048|0.8046|0.7405|0.5503|0.6014|0.3757|0.4805|0.4478|
>
> **Table:**  Training Loss Comparison of ViT-small with Different Initialization Methods on UCF Dataset Across Epochs
> ### Q3
> Limited Validation of Untrained Initialization Effectiveness.
> ### A3
> We demonstrate TAL's performance on unseen tasks in Table 5 of Section 5, encompassing tasks across diverse domains including Fashion MNIST (clothing and fashion items), FER2013 (facial expression recognition), and HAM10000 (dermatological skin lesion classification). The results indicate that models initialized with TAL consistently outperform those using random initialization and the LoGAH method.
> ### Q4
> Incomplete Comparison with Task Information.
> ### A4
> We augment the GHN-3 method with the same task information utilized in our TAL approach to evaluate its effectiveness. We initialize 12-layer ViT-Tiny and 12-layer ViT-Small using both standard GHN-3 and our enhanced GHN-3 with task information (GHN-3 w/ T.I.), then assessed initialization quality on the Decathlon datasets. Results demonstrate that incorporating task information also improve GHN-3's performance.
> | Model      | Method         | Airc. | C100  | DPed  | DTD   | GSTR  | OGle  | SVHN  | UCF   | Flwr  | Avg\_Acc |
> |------------|--------------|------|------|------|------|------|------|------|------|------|--------|
> | **12-Tiny** | GHN-3        | 3.24 | 35.19 | 87.72 | 6.86  | 89.12 | 0.06  | 10.00 | 3.43  | 10.88 | 27.39  |
> |            | GHN-3 w/ T.I. | 5.31 | 22.03 | 84.66 | 11.49 | 94.01 | 0.06  | 10.00 | 1.28  | 21.27 | 27.79 **(+0.40)** |
> | **12-Small** | GHN-3        | 2.64 | 5.55  | 84.30 | 7.23  | 84.39 | 0.06  | 10.00 | 1.74  | 9.51  | 22.82  |
> |            | GHN-3 w/ T.I. | 6.60 | 21.84 | 82.70 | 14.10 | 91.52 | 0.06  | 10.00 | 1.23  | 32.25 | 28.92 **(+6.10)** |
>
> **Table:** Performance of untrained models initialized with GHN-3 and GHN-3 with Task Initialization (GHN-3 w/ T.I.)
> ### Q5
> Insufficient Motivation for the Ancestry Model.
> ### A5
> The role of the ancestry model is clearly articulated in Section 4. It provides both the soft labels necessary for Stage 1 training and task-specific labeling information for multitask scenarios. The ablation experiments in Table 8 of Section 5 quantitatively demonstrate the ancestry model's significant contribution to performance, confirming its critical importance in enhancing parameter initialization quality.
> ### Q6
> Ambiguity in "Learngene" Concept.
> ### A6
> The essence of the Learngene framework lies in condensing critical knowledge from an ancestry model to initialize downstream models. Our implementation employs an encoder-decoder architecture, departing from previous Learngene methods that relied on manual parameter extraction and heuristic designs. Despite innovations in implementation techniques, our approach maintains the basic paradigm of Learngene while providing greater flexibility and adaptability. In our experimental analysis, we conduct a detailed ablation study on the ancestry model, clearly demonstrating its significant contribution to performance improvement.

---

### Decision · Program_Chairs · 2025-05-01

**Decision:**

Accept (poster)

**Comment:**

This paper proposes an approach for initializing parameters for specific tasks by predicting parameters.  In particular, the authors use an encoder-decoder model to transfer information across tasks.  Three experts reviewed this paper with the majority arguing for acceptance, with the lone dissent both saying they are somewhat unfamiliar with the topic and not providing an update to their review or a discussion post-rebuttal.  Most comments focused on clarity questions or requests for additional comparisons, which were provided in the rebuttal with most reviewers noting that they were sufficient to address their concerns.  As such, the AC finds insufficient justification to overturn the majority recommendation to accept, and the authors are encouraged to consider the comments made by reviewers to improve their paper.